# CrossNorm: On Normalization for Off-Policy TD Reinforcement Learning

## Abstract

Off-policy temporal difference (TD) methods are a powerful class of reinforcement learning (RL) algorithms. Intriguingly, deep off-policy TD algorithms are not commonly used in combination with feature normalization techniques, despite positive effects of normalization in other domains. We show that naive application of existing normalization techniques is indeed not effective, but that well-designed normalization improves optimization stability and removes the necessity of target networks. In particular, we introduce a normalization based on a mixture of on- and off-policy transitions, which we call cross-normalization. It can be regarded as an extension of batch normalization that re-centers data for two different distributions, as present in off-policy learning. Applied to DDPG and TD3, cross-normalization improves over the state of the art across a range of MuJoCo benchmark tasks.

## 1 Introduction

Data and feature normalization are well established techniques in supervised learning that reduce training times and increase the performance of deep networks (LeCun et al., 1998; Ioffe & Szegedy, 2015). Intriguingly, normalization is not very common in deep reinforcement learning.

In this paper, we first evaluate the existing and widely used batch normalization and layer normalization in the context of off-policy TD learning methods. Results improve only little over those without normalization and often are substantially worse. This is surprising, since according to experience in supervised learning, normalization should improve stability. In deep off-policy TD learning, rather target networks have been the crucial part to stabilize optimization (Mnih et al., 2015; Lillicrap et al., 2016). Interestingly, we find that layer normalization allows us to remove target networks completely and still ensure stable training. Nevertheless, the performance with layer normalization is on average inferior to the variant with target networks.

In contrast to supervised learning, in off-policy TD learning there is not a single data distribution, but two distributions: one due to actions in off-policy transitions, and one due to actions proposed by the current policy. Consequently, we introduce a new feature normalization scheme – cross-normalization. It is an adaptation of batch normalization that normalizes features based on a combination of these two datasets. Reliable statistics for the normalization are ensured by computing the running sufficient statistics. In contrast to previous normalization approaches, cross-normalization consistently improves performance over the target network baseline. Learning is faster and empirically stable without the use of target networks.

We demonstrate these effects for two popular, state-of-the-art off-policy learning methods: DDPG (Lillicrap et al., 2016) and TD3 (Fujimoto et al., 2018). Adding cross-normalization to both methods consistently improves their performance on multiple MuJoCo benchmarks.

The paper also empirically evaluates stability for cross-normalization in the context of linear function approximators. This study on simpler problems gives some intuitive insights how normalization by mean subtraction can help stabilize divergent problems.

## 2 Background and Related Work

Reinforcement learning considers the problem of an agent interacting with an environment. The time is divided into discrete time-steps $t$. At each step, a policy selects an action $a \in \mathcal{A}$ as a function of

the current state ($\pi : \mathcal{S} \to \mathcal{A}$). This results in a transition into the next environment state $s'$ and a reward $r$. The *return* is the discounted sum of rewards $R_t = \sum_{i=t}^{T} \gamma^{i-t} r_t$ with $\gamma \in [0, 1]$ being the discount factor that reduces the weighting of distant rewards. Reinforcement learning optimizes the parameters of a policy $\pi$ to maximize the expected return $J_t = \mathbb{E}_\pi[R_t]$.

During optimization, the policy constantly changes and the gathered experience from the past becomes off-policy. For sample-efficient training, it is important to be able to learn from such data, hence the focus of our paper is on off-policy learning.

Many RL algorithms estimate the expected return of a policy as a function of a state and an action, called the action-value function: $Q(s, a) = \mathbb{E}_\pi[R|s, a]$. In the Determinstic Policy Gradient (Silver et al., 2014) formulation used by algorithms like DDPG (Lillicrap et al., 2016), a deterministic policy (actor) network that produces actions as $a = \pi(s; \theta_\pi)$ can be trained by performing gradient ascent over a loss defined through the $\theta_Q$-parametrized critic:

$$\nabla_{\theta_\pi} J(\theta_\pi) = \mathbb{E}_\mu \left[ \nabla_a Q(s, a; \theta_Q)|_{a=\pi(s;\theta)} \nabla_{\theta_\pi} \pi(s) \right]$$

where $\mathbb{E}_\mu$ denotes that the expectation is taken over samples from an experience memory governed by an off-policy distribution $\mu$. The critic network can be optimized by sampling $(s, a, r, s')$ tuples from the experience replay memory and using TD learning (Sutton, 1988) to minimize:

$$L(\theta_Q) = \mathbb{E}_\mu \left[ (Q(s, a; \theta_Q) - r - \gamma Q(s', \pi(s'); \bar{\theta}_Q))^2 \right]$$

Here $\bar{\theta}_Q$ parametrizes the *target network*, a moving average that slowly tracks the critic network's parameters: $\bar{\theta}_Q \leftarrow \tau \theta_Q + (1 - \tau)\bar{\theta}_Q$, with $0 < \tau \ll 1$. DDPG also uses a target network for the actor. Target networks are updated every time an optimization step is taken during critic training. It is pointed out by Lillicrap et al. (2016) that "target networks are crucial" to train critics in a stable manner.

## 2.1 TARGET NETWORKS

Target networks are a prominent ingredient in Deep RL algorithms that use off-policy bootstrapping. However, it is argued that the delayed credit assignment slows down learning, and that removing this dependency would be valuable (Plappert et al., 2018).

There have been several attempts at stabilizing off-policy Deep RL. Focusing on discrete action spaces (and extending DQN), Durugkar & Stone (2018) try to modify the TD update rule to prevent over-generalization amongst states. Kim et al. (2019) replace the $max$ operator in DQN with a $mellowmax$ operator (Asadi & Littman, 2017) and demonstrate that the resulting updates are contractions, and provide encouraging empirical results. In continuous control, Achiam et al. (2019) use the Neural Tangent Kernel (Jacot et al., 2018) to derive a new TD algorithm that is empirically shown to be stable without target networks.

Unlike these approaches, in this paper we make no changes to the underlying TD algorithms beyond inserting normalization layers in the neural network. We show that a careful usage of normalization is sufficient to eliminate the need for target networks.

## 2.2 FEATURE NORMALIZATION

The most common feature normalization for deep networks in supervised learning is Batch Normalization (Ioffe & Szegedy, 2015). During training it uses the mean and variance moments from a single batch for normalization; during inference, normalization is based on fixed moments, which are moving averages computed during training. Batch normalization has been shown to significantly smooth the optimization landscape stabilizing gradient estimation and to increase the speed of training (Santurkar et al., 2018). Batch re-normalization (Ioffe, 2017) is a subsequent improvement of batch-normalization, which uses the moving averages of the mean and variance during *both* training and inference of the network.

Layer-normalization (Ba et al., 2016) normalizes over the features in a layer. This makes results independent of the batch size and enables the application to recurrent layers. Layer normalization has been applied to D4PG (Barth-Maron et al., 2018). In algorithms like PPO (Schulman et al., 2017) and HER (Andrychowicz et al., 2017) a normalization by running moments was applied to the observations as a data pre-processing step.

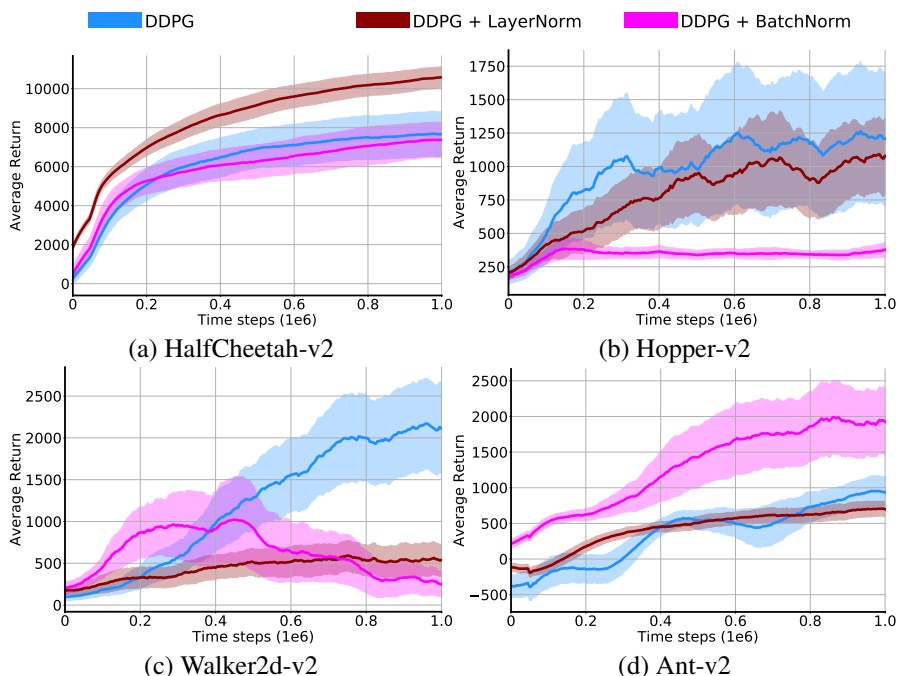

Figure 1: BatchNorm and LayerNorm applied to DDPG (all with target networks). Neither method improves performance consistently. Evaluation on OpenAI gym continuous control tasks, showing average returns and half standard deviations computed over 10 runs, curves are uniformly smoothed.

The original implementation of DDPG by Lillicrap et al. (2016) used batch normalization. However, it has not been widely used in DDPG implementations as direct application of batch normalization to off-policy learning is problematic. While training the critic, the action-value function is evaluated two times ($Q(s,a)$ and $Q(s',\pi(s'))$). Both $S$ and $S'$ come from the same distribution of states present in the experience replay. However, the *actions* come from different distributions: one is produced by the current policy $\pi(S')$ and the other was produced by previous policy iterations in the experience replay. This results in different mean features in the forward pass for $Q(s,a)$ and $Q(s',\pi(s'))$.

The dynamics of these distributions are very different: off-policy actions $a$ are expected to change very slowly, while the action distribution of the current policy $\pi(s')$ changes quickly during training. Batch normalization also distinguishes between a training and an evaluation mode. Using the evaluation mode for the target calculation would result in a normalization bias due to the different action distributions. At the same time using the training mode in the target calculation would result in different mean subtractions of the $Q$ function and its target.

The use of target networks in off-policy TD learning further complicates batch normalization. The target network's weights are a temporally delayed version of the training network resulting in feature distributions different from both of the above distributions.

## 2.3 BASELINE EXPERIMENTS

We evaluated the performance of layer and batch normalization in combination with DDPG on the standard continuous-control OpenAI gym MuJoCo tasks: *HalfCheetah*, *Hopper*, *Walker* and *Ant* (Brockman et al., 2016). We did our evaluations in the same way as Fujimoto et al. (2018). Both normalizations were applied to each layer of the critic after the activation function and to the observation layer, except in the case of LayerNorm where applying normalization to the input layer produces worse results. Batch normalization was used separately for $Q(s,a)$ and $Q(s',\pi(s'))$ in training mode. Other possible variants, like using the evaluation mode for the target computation, resulted in worse performance.

Figure 1 shows that LayerNorm works well on *HalfCheetah* and BatchNorm works well on *Ant*. This indicates that normalization can potentially be useful, however neither method produces consistent gains in performance over all four environments, as seen on *Walker*.

## 3 CROSS-NORMALIZATION

To address the problems of batch normalization in combination with Q-learning, we propose a simple new feature normalization strategy, which we call cross-normalization. It calculates the mean feature subtraction based on a mixture of features induced by on- and off-policy state-action pairs [1]. First, at each layer, the mean values for each feature of the critic network features are calculated over the batch[2]: $\mathbf{E}(f(s,a))$ and $\mathbf{E}(f(s',\pi(s')))$. Then a mixture of both mean values is used to normalize the features:

$$\hat{\mu}_\alpha = \alpha \cdot \mathbf{E}(f(s,a)) + (1-\alpha) \cdot \mathbf{E}(f(s',\pi(s'))) \tag{1}$$

The hyperparameter $\alpha$ determines the relative weighting of on- and off-policy samples. Fairly balancing both distributions ($\alpha = 0.5$) is an intuitive choice. In this case normalization can be applied by concatenating the two batches as $\tilde{s} = (s, s')$ and $\tilde{a} = (a, \pi(s'))$ and running a single forward pass of $(\tilde{s}, \tilde{a})$ tuples through the critic and using an existing BatchNorm implementation. Figure 2 shows that his approach consistently improves performance over all environments. Even though the training was performed without target networks, the normalization allowed for stable training. We call this variant **CrossNorm**.

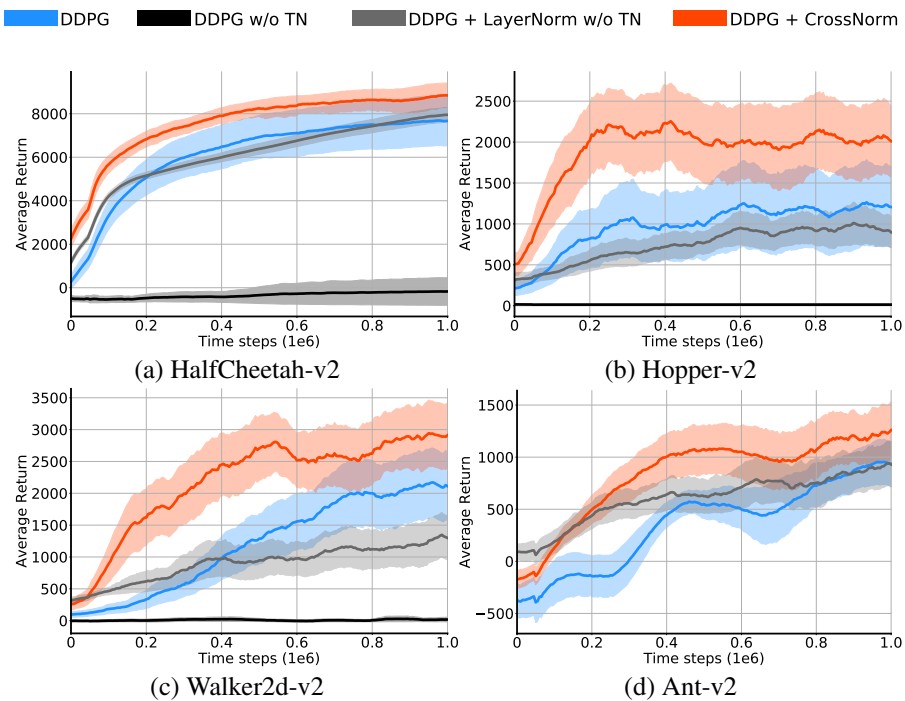

Figure 2: DDPG CrossNorm does not require target networks and outperforms other methods. Evaluation on OpenAI gym continuous control tasks, showing average returns and half standard deviations computed over 10 runs; curves are uniformly smoothed.

For TD3 this approach did not consistently improve performance, as seen in Figure 3. We believe this happens due to inaccurate mean and variance estimates. To verify this hypothesis we repeated the experiment with the same hyperparameters, but used a very large batchsize of 2048 to ensure precise mean and variance estimates. After the mean and variance were obtained from the large batch,

---

[1]Off-policy pairs are entierly from the replay-buffer $(s, a)$, whereas on-policy pairs make use of the current policy $(s', \pi(s'))$.

[2]The feature value indices are omitted to simplify the notation and $\hat{\mu}$ is a vector.

the training step was concluded with the original batchsize of 256 as in the previous CrossNorm experiment. The result is shown in Figure 3 in dark red. As expected, with the better normalization parameters, performance improved.

A large batchsize for the forward pass is considerably slower computationally. To continue using a smaller batch we apply two strategies that increase mean and variance estimate stability. Firstly we also consider unbalanced weightings of $\alpha$. As the distribution of the off-policy actions from the experience replay changes considerably slower than the action distribution of the constantly changing current policy, the mean features of the off-policy data are more stationary. Note that the mixing only applies to the mean features; the standard-deviation used for normalization is that of the entire joint batch. We find experimentally that $\alpha = 0.99$ produces the best results, as shown in Figure. 8. Secondly we apply batch re-normalization, which uses the running mean and variance values computed over several batches. We call this variant **CrossRenorm** and summarize it in Algorithm 1. It achieves similar performance as a large batchsize for the forward pass, but is considerably faster, making it the better option for feature normalization.

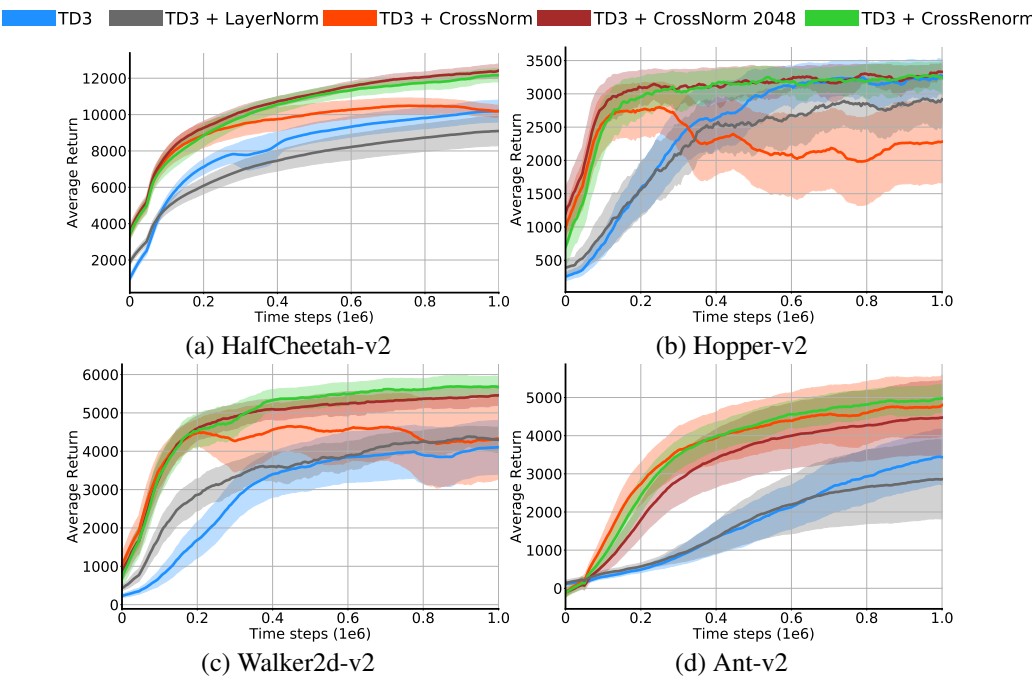

Figure 3: TD3 CrossRenorm outperforms TD3, all experiments are run with batch size 256. Evaluation on OpenAI gym continuous control tasks, showing average returns and half standard deviations computed over 10 runs; curves are uniformly smoothed.

## 3.1 RESULTS FOR DDPG AND TD3

The results for DDPG are shown in Figure 2. For CrossNorm the performance is improved across all four tasks, especially in the beginning. Moreover, the training is stable despite the missing target network. In comparison, the original DDPG without target network is not able to learn in any of the tasks. The best $\alpha$ value for CrossNorm was $\alpha = 0.5$. Following the success of training DDPG CrossNorm, which does not use target networks, we also tried training DDPG with LayerNorm without target networks. This combination resulted in stable training, a fact that is not widely known. This shows that that different normalization methods can be use to enable training without target networks. Again we see faster improvement, especially in the beginning.

Further we evaluated cross-normalization applied to the state-of-the-art algorithm TD3 (Fujimoto et al., 2018). TD3 is an improved version of DDPG, which accounts for the over-estimation bias by training two critics and by using the minimum of the predicted return. Again we removed the target network when applying cross-normalization.

**Algorithm 1:** Cross-Normalization pseudocode. CrossNorm uses $\alpha = 0.5$ and the BatchNorm function in line 5. CrossRenorm uses $\alpha = 0.99$ and the BatchRenorm function (i.e., normalizes with running averages of the moments).

**Input** : Off-policy transitions $(s, a, s')$, feature layer activations $f$, current policy $\pi$, mixing param. $\alpha$, and batch size $N$

**Output** : Normalized feature vectors $y$

1 $\mathbf{E}(x) := \frac{1}{N} \sum_i x_i$

2 $\hat{x}_{\text{off}} \leftarrow \mathbf{E}(f(s, a)); \quad \hat{x}_{\text{on}} \leftarrow \mathbf{E}(f(s', \pi(s')))$

3 $\hat{\mu}_\alpha \leftarrow \alpha \cdot \hat{x}_{off} + (1 - \alpha) \cdot \hat{x}_{\text{on}}$

4 $\hat{\sigma}^2 \leftarrow \frac{1}{2 \cdot N - 1} \left[ (\hat{x}_{\text{on}} - \hat{\mu}_{\alpha = 1/2})^2 + (\hat{x}_{\text{off}} - \hat{\mu}_{\alpha = 1/2})^2 \right]$

5 $y_{\text{on/off}} \leftarrow Normalize(x_{\text{on/off}}, \mu_\mathcal{B} = \hat{\mu}_\alpha, \sigma_\mathcal{B} = \hat{\sigma})$

The results are shown in Figure 3. The application of CrossNorm to TD3 did not produce good results and the sequence of steps by which we arrive at CrossRenorm is described in Section 3. BatchNorm may be failing because TD3 is learning more quickly. This makes normalization more difficult, as it needs to make precise estimates of more quickly moving distributions.

We test two strategies — CrossNorm and CrossRenorm — both of which ensure that the *same* moments are used to normalize the feature activations for the consecutive timesteps in a TD update. We find that a stable and stationary estimate of the moments, provided by CrossRenorm with are beneficial. On the MuJoCo tasks, it boosts the performance of methods like TD3 beyond the state-of-the-art achieved by TD3

and Soft Actor-Critic (SAC) as seen in Figure 4, while dropping the requirement of target networks. To the best of our knowledge, this is the first time a DDPG variant has been trained successfully without target networks.

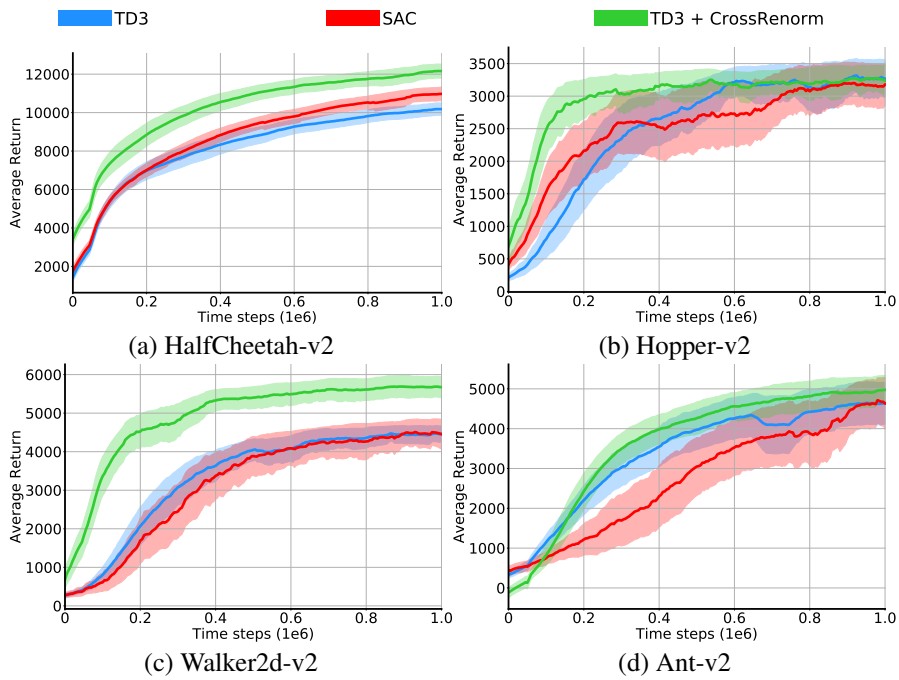

(a) HalfCheetah-v2

(b) Hopper-v2

(c) Walker2d-v2

(d) Ant-v2

Figure 4: TD3 CrossRenorm outperforms other baselines. Here we use the original hyperparameters by Fujimoto et al. (2018) and Haarnoja et al. (2018) re-evaluate on -v2 environments. Evaluation on OpenAI gym continuous control tasks, showing average returns and half standard deviations computed over 10 runs; curves are uniformly smoothed.

# 4 ANALYZING THE STABILITY IMPROVEMENT

In the previous section, we showed that normalization empirically provides improved stability without the use of target networks. What causes this effect? To isolate the cause that prevents divergence, we trained a DDPG agent on a fixed experience dataset with three different configurations: no target networks, with CrossNorm, and with a mean-only variant of CrossNorm. Figure 5 shows that mean-recentering in CrossNorm is enough to ensure stable optimization.

It is intriguing that mean-recentering provides stability. It motivated us to study the effects in a controlled setting: policy evaluation with simple linear function approximators.

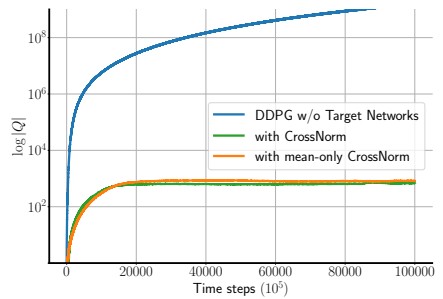

Figure 5: Plots showing the convergence of DDPG on a fixed experience buffer of *Walker* transitions with size 1 million. The curves are the log of the average Q value prediction of a randomly sampled $(s, a)$ batch of size 1024. All DDPG runs are trained without target networks.

## 4.1 EFFECT OF MEAN FEATURE SUBTRACTION

In the linear function approximator setting the value function for a state $s \in \mathcal{S}$ is computed as: $V(s; \theta) = \theta^\top \phi(s)$ or $Q(s, a; \theta) = \theta^\top \phi(s, a)$ where $\phi(s) \in \mathbb{R}^n$ is a feature vector of dimensionality $n$.

We consider the possible effect of subtracting the mean from the features of the Q function approximator. In a situation like off-policy DDPG-learning, the input of the Q-function is a combined representation (e.g. concatenated vectors) of a state and an action. During the TD update those are: $\phi = \phi(s, a)$ and $\phi' = \phi(s', \pi(s'))$. While both $s$ and $s'$ are drawn from the same distribution of available states in the experience replay, the actions come from two different distributions: $\mu$ and the current policy $\pi$. It is not clear which of the two distributions to use in order to calculate the mean features, so we defined in Equation 1 a parameterized mixture normalization with parameter $\alpha$ that uses both. Thus we use a combination of the means of the current ($\phi$) and successor ($\phi'$) features: $m = \mathbb{E}_\mu [\alpha \, \phi + (1 - \alpha) \, \phi']$. In practice we will only be able to calculate *estimates* of $m$ (e.g. by averaging a minibatch). Therefore, we also consider the stability of parameterizations, where $\alpha$ and $\beta$ do not sum to 1:

$$m = \mathbb{E}_\mu [\alpha \, \phi + \beta \, \phi'] \tag{2}$$

Recentering the feature encoding by $m$ gives us $\hat{\phi}(s, a) = \phi(s, a) - m$, which we then use as the new input features for the function approximator.

## 4.2 POLICY EVALUATION EXPERIMENTS

To test the stability of TD bootstrapping in isolation, we want to learn value functions for fixed policies. This configuration is called *policy evaluation* and can be run on fixed experience buffers (Sutton & Barto, 2018) that were generated by other policies, resulting in off-policy learning. While the training dynamics in policy evaluation differ from the more complicated concurrent learning of actor and critic, it provides a good indication of stability. Approximate off-policy learning suffers from the risk of divergence. A number of surprisingly simple MDPs exist that demonstrate divergence of off-policy TD(0) methods; one of these is *Baird's counter example* described in detail in Baird (1995); Sutton & Barto (2018).

We tested the effect of feature mean-recentering with different $\alpha$ and $\beta$ values on the following tasks a) Baird's counterexample, which has a linear function approximator, b) alterations of Baird's counterexample with randomly selected features and c) learning the value function of a Walker2d task (Brockman et al., 2016) from a fixed experience replay memory, with a neural network from which only the last layer is trained. The results are shown in Figure 6. For each task all the rewards are set to 0, therefore the true Q-value is 0 for every state. The mean absolute Q-value prediction is shown on a logarithmic scale. There are three regions visible in the diagram: an area of strong divergence (yellow), an area of fast convergence (blue), and an area of relatively slow convergence (green). We test for convergence through semi-gradient dynamic programming which simulates

the full expected TD update iteratively (Sutton & Barto, 2018). As expected, the case without normalization ($\alpha = \beta = 0$) is divergent in multiple tasks. Surprisingly, along the $\beta = 1 - \alpha$ line the policy evaluation converged for all $\beta > 0$ values. Also ignoring the distribution of the target Q-function features ($\alpha = 1$ and $\beta = 0$) is very close to the highly unstable region in all four cases. This indicates that even small deviations in the mean feature estimations could lead to divergence.

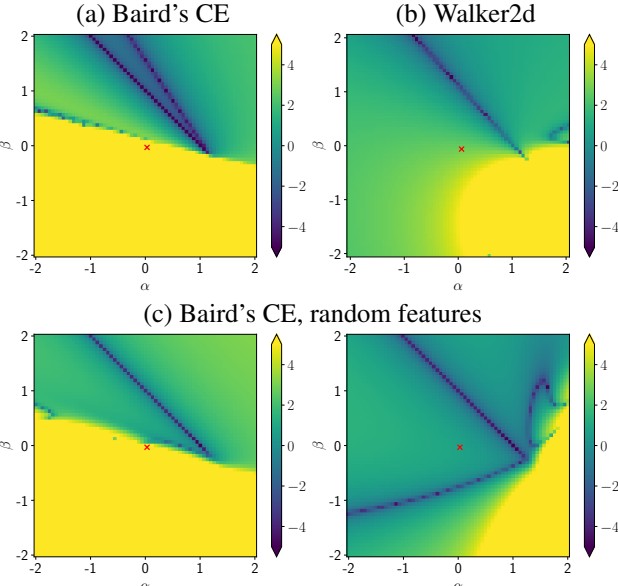

Figure 6: **(a)** and **(b)** are phase diagrams showing Baird's counterexample (CE) $\log(|\bar{V}|)$ and Walker2d's $\log(|\bar{Q}|)$ value estimates after optimization, on the $\alpha, \beta$ plane of the feature space. Lower values are better as all rewards were set to zero. $\alpha + \beta = 1$ normalization for $\alpha > 1$ produces stable results. Each pixel represents one optimization run for 50k iterations, with results produced using expected TD(0) updates with $\gamma = .99, \eta = 10^{-3}$. **(c)** shows two modifications of Baird's MDP with randomly generated feature vectors. The red cross indicates the un-normalized configuration.

In our experiments with a large number of randomly generated MDPs, we found that CrossNorm stabilized learning on all of them except for certain classes of contrived transition matrices. Therefore, we emphasize that CrossNorm does *not* provide convergence guarantees. However, we hypothesize that most MDPs of practical interest may be of a benign class that is amenable to CrossNorm.

## 5 CONCLUSION

We identified that normalization based on a mixture of on- and off-policy transitions is an effective strategy to mitigate divergence and to improve returns in deep off-policy TD learning. The proposed cross-normalization methods are modular modifications of the function approximator. Thus, they can be applied to DDPG, TD3, and potentially other off-policy TD algorithms. For both tested algorithms, cross-normalization stabilized training and improved the results in terms of the reward achieved. Moreover, it increased stability sufficiently to enable training without target networks. Further experiments have shown that mean-only normalization is sufficient to stabilize training. We also studied the effects of normalization in more controlled settings. Different mixtures of on- and off-policy normalization result in a structured space of stable solutions. Cross-normalization increases the chance to hit the most stable areas.

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

## A  APPENDIX

**Hyper-Parameters**  We used the well tuned "OurDDPG" and TD3 code published by Fujimoto et al. (2018) to produce the respective DDPG and TD3 baseline curves. All of our CrossNorm improvements were small modifications to these files. The only change to the network architectures was the incorporation of normalization layers after the nonlinearity in each layer and on top of the input layer (as was done in the original DDPG paper by Lillicrap et al. (2016)). After that, we performed concatenated forward passes through the critics in the Cross Normalization variants.

To obtain the final set of hyperparameters, we manually tried out various combinations of different learning rates for the actor and critic, and chose between the RMSprop and Adam optimizers. We found that when training without target networks, RMSprop performed better than Adam.

**Experiment details**  We implemented a custom CrossNorm layer in PyTorch with configurable $\alpha, \beta$ and an option to switch on renormalization.

For CrossNorm experiments with $\alpha = 0.5$, we used BatchNorm layers with concatenated forward passes to produce the consecutive Q predictions. Our code used the BatchNorm layer provided in PyTorch which relies on a highly optimized C++ implementation which runs about twice as fast in wall-clock time as our custom CrossNorm layer. The BatchNorm momentum argument was set to 1 (which should correspond to 0 in TensorFlow).

For CrossRenorm, we used our custom CrossNorm layer with $\alpha = 0.99, \beta = 0.01$. After 5000 optimization steps, we switched on normalization by running averages as in renormalization. Like in the TD3 paper, before executing the trained policy we perform 1000 timesteps of random exploration in the Walker and Hopper environments, and 10000 steps in HalfCheetah and Ant.

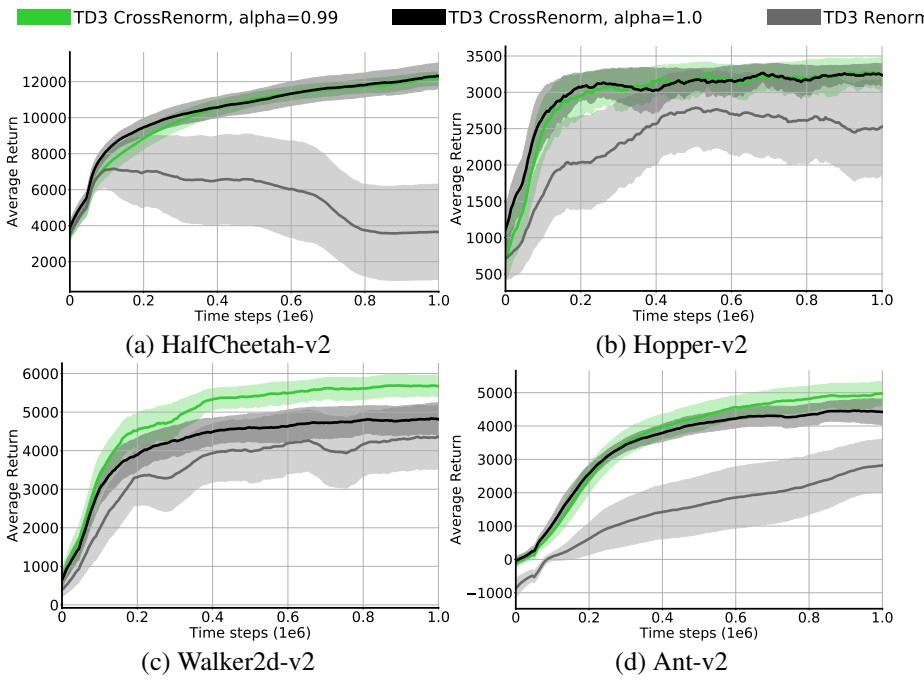

Figure 7: TD3 CrossRenorm comparison between $\alpha = 0.99$ and $\alpha = 1.0$ and batch re-normalization (without target networks). All curves are smoothed averages over 10 seeds.

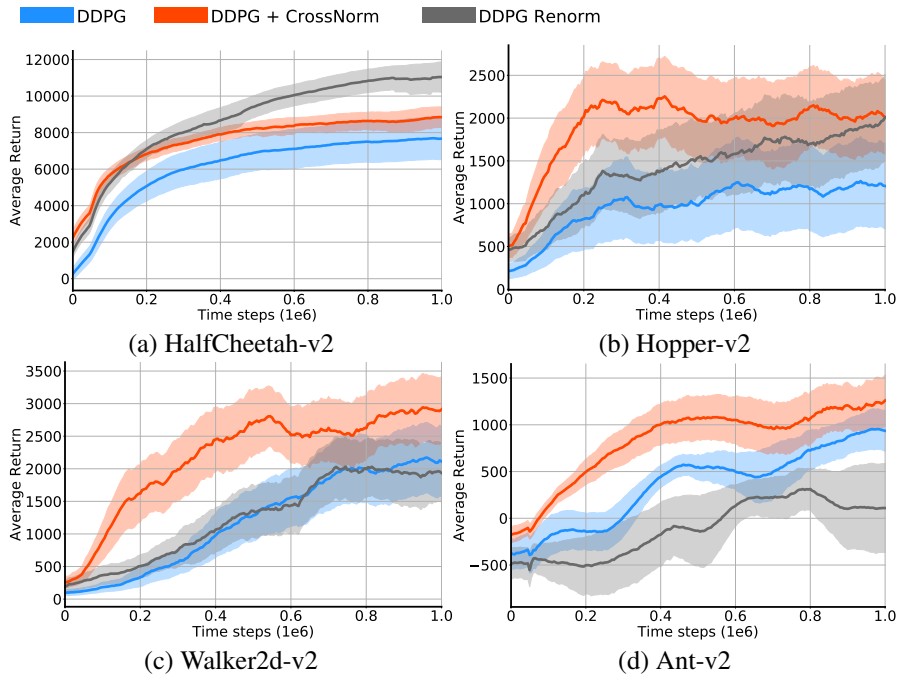

Figure 8: Comparison to DDPG with batch re-normalization (without target networks). All curves are smoothed averages over 10 seeds.

**Usage suggestions** As is common in RL experimentation, every algorithm requires careful hyper-parameter adjustment. The same must be done when using CrossNorm / CrossRenorm. Here we provide advice for various possible issues that could arise:

- **Learning rates**: When training without target networks, care must be taken to not use a very high learning rate for the critic, otherwise the estimated Q values could diverge. In general there is always an upper bound to the usable learning rates in RL; when using high learning rates it is more appropriate to use RMSprop as the optimizer as it never increases (only decreases) the step size beyond the specified one. With Adam, it is safer to use smaller base learning rates because Adam can also increase the step size.
  The scale of the reward also affects the effective step size, and we found that in the SURREAL tasks, it was useful to have the rewards scaled to similar ranges as the four MuJoCo tasks, to ensure fast learning.

- **Performance collapse**: TD3, while very efficient, can be sensitive to normalization. A common pathology could be that after achieving high returns for a while, predicted Q values start collapsing to smaller values, which is possibly related to the underestimation bias induced by the clipped double-Q estimator. We found that using large batch sizes, or instead using CrossRenorm, made the training robust against this problem. With CrossRenorm, it is advisable to experiment with different start-times for switching on the renormalization. We found 5000 iterations to be a good setting.

- **Low maximum performance**: In general, the benefit of CrossNorm is in making optimization faster. This often translates to good learning performance, but sometimes it does not. That is typically caused by a lack of adequate exploration — which is not addressed by normalization — which could happen if the agent learns so fast that it gets stuck in a bad local optimum before it has had the chance to explore sufficiently. In these cases, it is important to remember that using CrossNorm could change the optimal values of other unrelated hyperparameters (as in Figure 9b), which would need to be adjusted to get the best performance.

## A.1 ADDITIONAL ENVIRONMENTS

To further demonstrate the performance of CrossNorm we evaluate the method on three additional environments. The first of these is the Humanoid form the OpenAI gym, this environment is more complex in having a higher dimensional observation space. The second two are robotics manipulation environments from the SURREAL paper Fan et al. (2018) called SawyerLift and BaxterPegInHole. The results are shown in Figure 9. In these experiments we found that our normalization approach helps increase performance over the TD3 baseline in two out of three cases. Especially the robotics tasks are dependent on good exploration which is not addressed by our method which only improves optimization performance.

For Humanoid, we used a learning rate of $5 \cdot 10^{-4}$ for both the actor and the critic in all configurations. For the SURREAL tasks, we used a reward scaling of 10 and a learning rate of $1 \cdot 10^{-4}$ for both the actor and the critic networks.

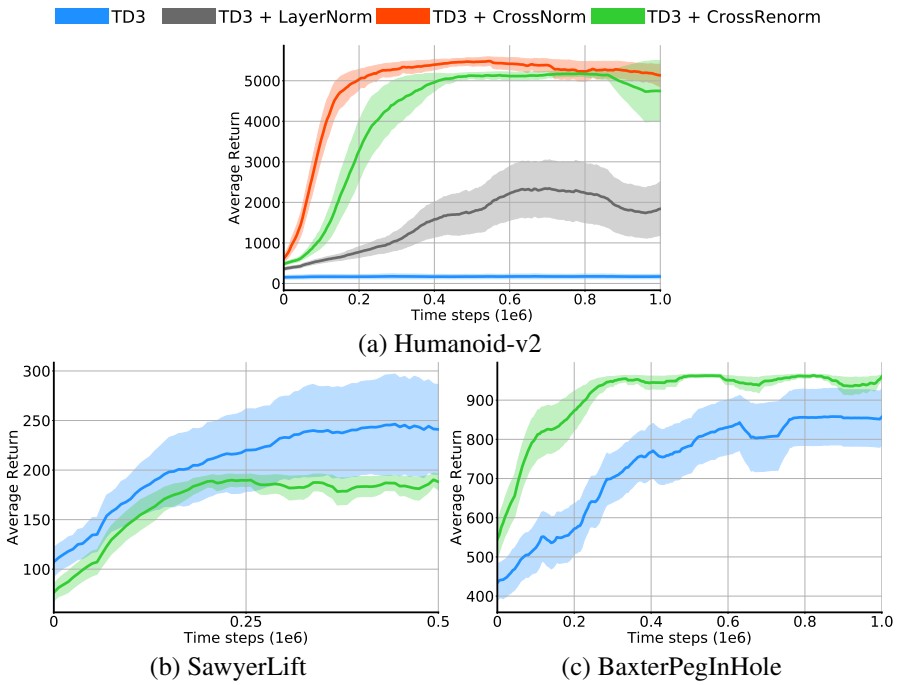

Figure 9: Additional Experiments showing performance of CrossNorm on a more complex gym task and two robotic manipulation environments. The Humanoid curves are smoothed averages over 10 random seeds. The SURREAL experiments are smoothed averages over 4 seeds each.

Table 1: List of hyperparameters of algorithms used in the Paper. The remaining hyperparameters were the same as in Fujimoto et al. (2018). TD3 with CrossNorm 2048 used a forward pass of batch size 2048 only to get the mean and variance of the batch for normalization, while the training was conducted with batch size 256.

| Algorithm Name | Fig. | LR | $\tau$ | Batch Size | Optimizer | $\alpha$ |
|---|---|---|---|---|---|---|
| DDPG | 1 | $10^{-3}$ | $5 \cdot 10^{-3}$ | 100 | Adam | - |
| DDPG with LayerNorm | 1 | $10^{-3}$ | $5 \cdot 10^{-3}$ | 100 | Adam | - |
| DDPG with BatchNorm | 1 | $10^{-4}$ | $5 \cdot 10^{-3}$ | 100 | RMSprop | - |
| DDPG | 2 | $10^{-3}$ | $5 \cdot 10^{-3}$ | 100 | Adam | - |
| DDPG w/o TN | 2 | $10^{-3}$ | - | 100 | Adam | - |
| DDPG w/o TN w/ LayerNorm | 2 | $10^{-3}$ | - | 100 | Adam | - |
| DDPG CrossNorm | 2 | $10^{-4}$ | - | 100 | RMSprop | 0.5 |
| TD3 | 3 | $10^{-3}$ | $5 \cdot 10^{-3}$ | 256 | Adam | - |
| TD3 w/o TN w/ LayerNorm | 3 | $10^{-3}$ | - | 256 | Adam | - |
| TD3 CrossNorm | 3 | $10^{-3}$ | - | 256 | RMSprop | 0.5 |
| TD3 CrossNorm 2048 | 3 | $10^{-3}$ | - | 256 (2048) | RMSprop | 0.5 |
| TD3 CrossRenorm | 3 | $10^{-3}$ | - | 256 | RMSprop | 0.99 |
| TD3 | 4 | $10^{-3}$ | $5 \cdot 10^{-3}$ | 100 | Adam | - |
| SAC | 4 | $10^{-3}$ | $5 \cdot 10^{-3}$ | 256 | Adam | - |
| TD3 CrossRenorm | 4 | $10^{-3}$ | - | 256 | RMSprop | 0.99 |

