# OpenReview forum: "CrossNorm: On Normalization for Off-Policy Reinforcement Learning"
_ICLR.cc/2020/Conference — Reject_

### Official Review · AnonReviewer2 · 2019-10-23
**Official Blind Review #2**

**Rating:** 3

**Review:**

This paper studies the problem of feature normalization in off-policy RL, more specifically, learning a Q function with continuous action from off-policy data. It shows standard feature normalization methods in supervised learning is indeed not effective for RL settings, due to the fact that a and a’ are from very different distributions with different dynamics. Since the batch of a and a’ come to model iteratively, standard normalization method suffers from this 2-periodic distribution shift. This paper proposes a normalization method, by merging a and a’ into a * single* update step to the batch normalization layer.

This paper does catch a problem and uses a straightforward solution but empirically effective, but I still have two main concerns. 1) This paper only shows benefit 4 tasks in the MoJoCo domain. 2) As the solution is relatively straightforward, with very restrictive applicable settings (particular normalization trick with particular function approximator). It’s less clear to me whether this provides enough contribution and inspiration to other work as a conference paper. I tend to vote for reject at this time.

I would like to first point out the pros of this paper from my perspective then explain my main concerns point by point. This paper does a great job of capturing the dilemma of batch normalization in RL settings. My understanding is that the problem is caused by a periodic distribution shift between a and a’. Because we have to pass a and a’ in separate batch and BatchNorm does an online updating after each batch, we are in a dilemma, as the paper pointed out. If we don’t update BatchNorm in one of them (e.g. target value) that will be biased and make BatchNorm ineffective, and if we do so we will have a systematic difference between Q(s, a) and Q(s’, a’).

Main concerns:
1) This paper only shows benefit 4 tasks in the MoJoCo domain. Given that the empirical result is pretty much the only support of the claim in this paper, the lack of more diverse experiments would weaken the contribution.

2) The dilemma is totally caused by that BatchNorm will immediately perform update according to a batch after it is input, then a lazy update will cancel this: do a single update to the BatchNorm layers, for two batches of data (a, a’). This is equivalent to the proposed solution when alpha=0.5. It needs not to be a weakness for the algorithm itself as we appreciate simple but effective algorithm. However this makes the problem itself more like a design weakness of BatchNorm and a simple patch to fix it. I doubt how much algorithmic insight this paper could contribute, to inspire related research.
Minor point: It also makes me doubt whether we really need an alpha or not.

3) As the paper pointed out “using the training mode in the target calculation would result in different mean subtractions of the Q function and its target.” This means it will have a systematic difference in Q function used to compute Q(s, a) and Q(s’, a’), but isn’t this also true for the target network since we are using a different network to compute Q(s’, a’). Eventually, if the policy converges, those differences will disappear. So why target network will have no problem but this will. In general, I’d like to see a more clear analysis about the dilemma of BatchNorm in off-policy data, and why the two simple ways won’t work.


**Experience Assessment:**

I have published one or two papers in this area.

**Review Assessment: Checking Correctness Of Derivations And Theory:**

N/A

**Review Assessment: Checking Correctness Of Experiments:**

I assessed the sensibility of the experiments.

**Review Assessment: Thoroughness In Paper Reading:**

I read the paper at least twice and used my best judgement in assessing the paper.

---

> ### Author Response · Authors · 2019-11-13
> **Response**
>
> We thank the reviewer for their valuable comments and useful suggestions.
>
> > This paper only shows benefit 4 tasks in the MoJoCo domain.
>
> This is a valid concern; therefore, we have added more experiments to the paper.
>
>  In addition to the four MuJoCo tasks and our experiments with linear function approximators  we have also successfully tested CrossNorm on non-locomotion  SURREAL/robotsuite tasks, and the much harder gym humanoid task. These are described in the "Additional Experiments" section added to the appendix. Further, we also successfully applied CrossNorm to achieve top-10 placement in the NeurIPS 2018 AI for Prosthetics challenge. (Not specified further to avoid breaking anonymity.)
>
>
> > As the solution is relatively straightforward, with very restrictive applicable settings (particular normalization trick with particular function approximator). It’s less clear to me whether this provides enough contribution and inspiration to other work as a conference paper.
>
> We regard the simplicity of the method as an advantage. The method is applicable to all DPG-style approaches using deep neural networks as function approximators, which is a major area of continuous-control Deep RL research.
>
> Using Target Networks (TNs) in Deep RL is an empirically motivated algorithmic fix. It changes the original TD learning algorithm into a two-timescale algorithm; TD learning methods do not generally have polyak averaging of parameters, and it is not understood how using a TN provides stability. In fact, there is evidence that TNs do not avoid divergence, but merely delay it [1],
>
> Prevention of divergence without target networks has been an open problem that has seen much attention [2][3][4]. All of these approaches, while valuable and interesting, are *new algorithms* with special loss functions and update rules that do not perform the original DPG-style off-policy actor-critic training. Our demonstration, on the other hand, is significant because it is the simplest approach of them all - simply augmenting the function approximator class with BatchNorm/LayerNorm - and it works just fine with the old algorithms. We believe this is an interesting and surprising result.
>
> Batch normalization is not a trick but widespread in supervised learning. Being rarely used in RL, it is important to show how to apply normalization properly, as it provides substantial performance improvements. Moreover, we demonstrate that LayerNorm, which is being used by several DDPG implementations [e.g. OpenAI Baselines], allows stable training without target networks, which has not been shown before.
>
> [1] https://arxiv.org/abs/1812.02648
> [2] https://arxiv.org/abs/1903.08894
> [3] https://openreview.net/forum?id=Bk-ofQZRb
> [4] https://www.ijcai.org/proceedings/2019/0379.pdf
>
>
> > The dilemma is totally caused by that BatchNorm... It needs not to be a weakness for the algorithm itself as we appreciate simple but effective algorithm. However this makes the problem itself more like a design weakness of BatchNorm and a simple patch to fix it. I doubt how much algorithmic insight this paper could contribute, to inspire related research ...
>
> Joining two batches and doing a joint forward pass through BatchNorm is how we implement the alpha = 0.5 case (we describe it in the paper). However, we found that TD3 is particularly sensitive to the details of normalization (shown in Figure 3) and empirical results for alpha = .99  were better with renormalization.
>
> We think that our paper yields the following insight: in the set of deep function approximators, there could exist function classes (such as those containing LayerNorm / BatchNorm layers) with which off-policy TD learning is stable under certain additional assumptions. While a theoretical proof may be difficult, this insight gives others another tool to improve stability.
>
> >  As the paper pointed out “using the training mode in the target calculation would result in different mean subtractions of the Q function and its target.” This means it will have a systematic difference in Q function used to compute Q(s, a) and Q(s’, a’), ... So why target network will have no problem but this will. In general, I’d like to see a more clear analysis about the dilemma of BatchNorm in off-policy data, and why the two simple ways won’t work.
>
> It is true that for target networks there is a systematic difference in the Q function used to compute Q(s, a) and Q(s’, a’). This does not prevent training because this is part of the algorithm’s design. There is also a difference in Q functions as result of normalizing with different data statistics (s,a), (s’,a’), which causes more problems. It is difficult to compare these differences to one another because these methods are also effectively different algorithms, in fact the Q functions *are required* to be different as part of the design of the TN algorithm.
>
>
> We hope this has addressed all of your concerns.

---

### Official Review · AnonReviewer1 · 2019-10-23
**Official Blind Review #1**

**Rating:** 6

**Review:**

The paper describes a novel normalization strategy for Off-Policy TD Reinforcement learning. Normally, Off-Policy TD RL is stabilized by usage of a target network, which has the disadvantage of slowing down the learning process. The paper first shows the effects of using existing normalization methods (batch and layer normalization) in the context of OPTD methods. Those approaches are shown to be inferior to target networks, because the data (actions in off-policy transitions and on-policy transitions) is coming from two different distributions. Experiments show that those normalization methods do not lead to consistent improvements over the benchmarks.
To tackle this problem, the authors introduce a cross-normalization scheme that works across the two datasets in this context. Cross-normalization is achieved by calculating a mixture of the mean values of both on- and off-policy state-action pairs. The weight of the contribution of those distributions is handled by a hyperparameter, which defaults to 0.5; thus using a balanced influence on both distributions (CrossNorm). Since the mean features of the off-policy data are more stationary, two strategies are applied: First, the hyperparameter is set to 0.99, thus giving more weight to the off-policy data, which is less volatile. Second, mean and variances are computed over several batches to increase stability (CrossRenorm). It finally is shown that the CrossRenorm approach is able to surpass state-of-the-art performance on the MuJoCo benchmark while having the benefit of not needing a target-network. In further experiments, it is shown that CrossNorm stabilizes learning in most contexts, but does not guarantee to converge in all settings.

Overall, the paper manages in a very clear and structured manner, (1) to show the current approaches for stabilizing learning and their downsides, (2) to show why common normalization methods fail and (3) formulates a possible solution for this problem. Furthermore, empirical results are not only shown, but also analysed.


**Experience Assessment:**

I do not know much about this area.

**Review Assessment: Checking Correctness Of Derivations And Theory:**

I assessed the sensibility of the derivations and theory.

**Review Assessment: Checking Correctness Of Experiments:**

I assessed the sensibility of the experiments.

**Review Assessment: Thoroughness In Paper Reading:**

I read the paper at least twice and used my best judgement in assessing the paper.

---

> ### Author Response · Authors · 2019-11-13
> **Response**
>
> We thank the reviewer for the positive feedback on our work.

---

### Official Review · AnonReviewer3 · 2019-10-23
**Official Blind Review #3**

**Rating:** 3

**Review:**

The paper introduces a new normalization scheme, cross normalization, that stabilizes the off-policy reinforcement learning algorithm. The results show that by simply performing batch-normalization, where the mean and variance statistics are computed with both behavior and target action samples, it can increase the performance of DDPG and TD3 algorithm consistently. The paper also shows that it prevents the algorithm from diverging even when the target network is removed, showing the source of stabilization.

The results are surprisingly good when the simplicity of the algorithm is considered. Nevertheless, I think the paper is not providing enough theoretical backups for the claimed algorithm, and it prevents me from being completely convinced. Also, the paper does not seem to be a complete draft - there are many points that seem to be incomplete. I think it would be much better if the paper develops some theory behind the normalization, referring some previous results as (Liu, Yao, et al. "Representation balancing mdps for off-policy policy evaluation." Advances in Neural Information Processing Systems. 2018.). For now, I feel that the paper is not ready for publication.

Here are some problems with the paper I found:

1. In the introduction, the paper says that the paper investigates convergence: where is the convergence investigation?

2. At the top of page 4, the sentence is not complete

3. Below eq (1), it is written as "the second order moments of the variance". Is it the second moment, or the variance? How do you convex combinate variances? Is it OK to do so?

4. While it is claimed in the paper that TD3 + CrossRenorm (alpha=0.99) performs well, it is not really justified why (alpha=0.99) is crucial. While it is written in the paper that " As the distribution of the off-policy actions from
the experience replay changes considerably slower than the action distribution of the constantly
changing current policy," such point can also be applied to DDPG and it does not explain why alpha=0.99 is needed for the only TD3. The paper also lacks experiments about BatchRenorms on DDPG and TD3, which would be a fair comparison against CrossRenorm.

5. Why do we need Figure 4? Is it only for the comparison against SAC?

6. Below eq (2), the paper says about big \Phi, but it is never defined and not used anymore. What is it about?

7. The stability improvement analysis implies that the mean-only crossnorm is sufficient for stabilization. Why do we need variance normalization then?





**Experience Assessment:**

I have read many papers in this area.

**Review Assessment: Checking Correctness Of Derivations And Theory:**

I assessed the sensibility of the derivations and theory.

**Review Assessment: Checking Correctness Of Experiments:**

I assessed the sensibility of the experiments.

**Review Assessment: Thoroughness In Paper Reading:**

I read the paper at least twice and used my best judgement in assessing the paper.

---

> ### Author Response · Authors · 2019-11-13
> **Response**
>
> We thank the reviewer for the constructive feedback, and for pointing out that our “results are surprisingly good when the simplicity of the algorithm is considered”.
>
>
> > I think it would be much better if the paper develops some theory behind the normalization
>
> There are theoretical studies and empirical studies. Both can be valuable. This paper is an empirical study. In fact, we tried to find a theoretical proof for stability in the simpler case of linear function approximators. However, we found counterexamples, which proves that there cannot be a proof for stability without additional assumptions. Nonetheless, the empirical study shows clear benefits of the approach, which makes it valuable. In Deep learning, the mechanism by which techniques like batch normalization and layer normalization improve optimization is still an open area of investigation [1][2].
> [1] https://papers.nips.cc/paper/7996-understanding-batch-normalization.pdf
> [2] https://papers.nips.cc/paper/7515-how-does-batch-normalization-help-optimization.pdf
>
>
> > In the introduction, the paper says that the paper investigates convergence: where is the convergence investigation?
>
> This sentence was talking about the linear function approximator case, it has been changed to “empirically evaluates stability”.
>
>
> >  Below eq (1), it is written as "the second order moments of the variance". Is it the second moment, or the variance? How do you convex combinate variances? Is it OK to do so?
>
> Thank you for pointing this out! That was a mistake in our writing and we have removed it. We do not convex-combinate the variances: the variance of the concatenated minibatch is computed as a whole (this happens automatically via the joint forward pass, with no special code on our part).
>
>
> >  While it is claimed in the paper that TD3 + CrossRenorm (alpha=0.99) performs well, it is not really justified why (alpha=0.99) is crucial. While it is written in the paper that " As the distribution of the off-policy actions from the experience replay changes considerably slower than the action distribution of the constantly changing current policy," such point can also be applied to DDPG and it does not explain why alpha=0.99 is needed for the only TD3.
>
> It is not “crucial” to use alpha=0.99. As shown by the comparison with the alpha=0.5 runs, we found that TD3 is particularly sensitive to the details of normalization in a way that DDPG is not. This is why we focused the effort on TD3 and chose to tune the mixing with alpha and empirically found .99 to be appropriate with renormalization.
>
> >  The paper also lacks experiments about BatchRenorms on DDPG and TD3, which would be a fair comparison against CrossRenorm.
>
> We evaluated BatchRenorm on DDPG and TD3 and have included these results in the supplemental material. Neither of these variants worked consistently better than the baselines already included. TD3 + BatchRenorm was often not stable.
>
>
> >  Why do we need Figure 4? Is it only for the comparison against SAC?
>
> Fig. 4 compares our modifications against the author-reported original hyperparameters of TD3 (and also SAC). Also the original hyper-parameters of TD3 (batch size=100) are different compared to Fig. 3 (batch size=256), giving it slightly better performance.
>
>
> > Below eq (2), the paper says about big \Phi, but it is never defined and not used anymore. What is it about?
>
> This describes how we normalize in the linear function approximator case, it has been clarified in the paper by replacing with \phi.
>
>
> > The stability improvement analysis implies that the mean-only crossnorm is sufficient for stabilization. Why do we need variance normalization then?
>
> The mean-only crossnorm is only required to avert divergence, however the additional variance normalization significantly improves the training speed.
>
>
> We hope that these additional experiments and clarifications have improved the paper.

---

### Author Response · Authors · 2019-11-13
**General Response**

We thank the reviewers for the feedback and the detailed reviews. We have incorporated these suggestions to improve our paper.

To address the main point of concern: the applicability of the method to a diverse set of environments, we did additional experiments with the gym humanoid environment and two SURREAL/robosuite robotic manipulation tasks, both now shown in the supplemental material. Further we added comparison to normal batch renormalization to the supplemental material.

We address the questions and comments of the reviewers below.

---

### Decision · Program_Chairs · 2019-12-19

**Decision:**

Reject

**Comment:**

This is certainly a boarderline paper. The reviewers agreed this paper provides a good explanation and empirical justification of why popular normalization schemes don't help in DRL. The paper then proposes a simple scheme and demonstrates how it improves learning in several domains. The main concerns are the nature of these gains and how broadly useful the new approach is. In many cases there appear to be somewhat clear wins in the middle of the learning curves, but by the end of each experiment the errorbars overlap. The most clear results are those with TD3. There are some oddities here: using half SD error bars and smoothing---both underline the concern about significance.

The reviewers requested more experiments and the authors provided three more domains: two in which their method appears better. These are not widely used benchmarks and it was hard to compare the performance of the baselines with fan et al (different setup) to evaluate the claims. The paper nicely provides lots of insight and empirical wisdom in the appendix, explaining how they got the algorithms to perform well.